# Factors influencing physical distancing compliance among young adults during COVID-19 pandemic in Indonesia: A photovoice mixed methods study

**Ahmad Junaedi**[1], **Ken Ing Cherng Ong**[1]*, **Fauzan Rachmatullah**[2], **Akira Shibanuma**[1], **Junko Kiriya**[1], **Masamine Jimba**[1]

1 Department of Community and Global Health, Graduate School of Medicine, The University of Tokyo, Bunkyo-ku, Tokyo, Japan, 2 School of Health and Related Research, University of Sheffield, Sheffield City Centre, United Kingdom

* kenicong@m.u-tokyo.ac.jp

**Data Availability Statement:** All data are available at https://data.mendeley.com/datasets/9bb4s4tz75/1.

## Abstract

The spreading of the coronavirus disease (COVID-19) is growing out of control in Indonesia since the first two confirmed cases were announced in March 2020. Physical distancing measures are key to slowing down COVID-19 transmission. This study investigated factors associated with physical distancing compliance among young adults in the Jakarta Metropolitan Area, Indonesia. A convergent photovoice mixed methods design was used. Quantitatively, using data from 330 young adults in Jakarta Metropolitan Area, Indonesia, physical distancing compliance scores and its associated factors were analyzed with hierarchical linear regression. Responses from 18 young adults in online focus group discussions and 29 young adults in photovoice were analyzed with thematic analysis. Then, the findings were integrated using joint displays. The mean compliance score of young adults was 23.2 out of 27.0. The physical distancing compliance score was higher among those who worked or studied from home ($\beta = 0.14$, $p < 0.05$), compared with those who resumed work at an office or study at school. Celebrating religious days ($\beta = -0.15$, $p < 0.05$) and having hometown in the Jakarta Metropolitan Area ($\beta = -0.12$, $p < 0.05$) were negatively associated with higher physical distancing compliance scores. Joint displays expanded the reasons for workplace policy, awareness, and social pressure as facilitators and barriers to compliance. Young adults' physical distancing compliance scores were high, but they are at risk of not complying due to religious events and changes in workplace policies. Beyond individual efforts, external factors, such as workplace policies and social pressure, play a major role to influence their physical distancing compliance.

## Introduction

A year after the first identification of the coronavirus disease (COVID-19), the virus continues to spread globally. As of September 1, 2021, more than 216 million cases and 4.5 million

**Funding:** This study was supported by The University of Tokyo Research Fund granted to MJ. The funder has no role in designing the study, collecting, analyzing and interpreting the data, preparing the manuscript, or decision to publish.

**Competing interests:** The authors declare no competing interests.

fatalities were documented in 192 countries/regions [1]. COVID-19 vaccination is insufficient to tackle this global threat. A combination of biomedical approaches with vaccination and social approaches might lead to successful prevention of the spread of the novel coronavirus [2]. Social approaches, such as physical distancing, as part of a non-pharmaceutical intervention (NPI), is an important measure to slow down virus transmission.

Physical distancing is one of the immediate responses to the COVID-19 pandemic. Physical distancing can reduce the total number of contacts and hence the chance of transmission [3–5]. At least 186 countries have introduced various levels of restrictions that include physical distancing [6]. Cooperation from community members is needed to achieve the goal of reducing the number of contacts.

Although physical distancing is unique to the current COVID-19 pandemic, evidence on the facilitators and barriers to adherence is still scarce. Many studies in past pandemics, such as H1N1 influenza in 2009, have focused on quarantine adherence [7, 8]. These studies showed that several factors were associated with quarantine adherence such as social and psychological factors, knowledge of the disease, cultural value, characteristics of the quarantine, and quarantine support characteristics [7–10]. In addition, policies at the workplace, such as leave entitlement, were a facilitator of quarantine adherence in the previous swine flu pandemic [8]. Moreover, the influence of the society affected reported compliance with the quarantine during the Severe Acute Respiratory Syndrome (SARS) outbreak in Canada [11]. However, facilitators and barriers to adequate public responses to the outbreak are not recognized in the emergency restriction guidelines [12].

The spreading of the COVID-19 infection is not under control in Indonesia. As of September 1, 2021, COVID-19 infection has been reported in every province with a total of more than 4.1 million cases and 133,000 deaths nationwide [13]. Among those infected, 24.8% of confirmed cases were among the 19–30 age group [13]. The Indonesian government responded to this situation by introducing physical distancing regulations as a part of the large-scale social restriction (called "*Pembatasan Sosial Berskala Besar*/PSBB" in Indonesian). Although several physical distancing rules have been implemented and extended multiple times, Indonesia still struggles to manage the community's compliance. Several mass religious gatherings resulted in COVID-19 transmission clusters [14, 15]. Moreover, evidence on the facilitators and barriers for physical distancing compliance in Indonesia is not available.

The risk of local transmission is high in Jakarta. Jakarta is the second-largest metropolitan area in Asia after the Tokyo Metropolitan Area, and 38.6% of the workforce are young adults who commute daily in this city [16, 17]. Moreover, as Indonesia has 633 ethnic groups and 6 official religions across the archipelago, various cultural and religious activities take place from time to time that can make it difficult for people to keep physical distance with others [18, 19]. This study was conducted to identify facilitators and barriers to compliance with physical distancing measures among young adults in the Jakarta Metropolitan Area.

## Methods

### Study design and settings

In this convergent mixed methods cross-sectional study, data were collected through quantitative online self-administered questionnaires and qualitative photovoice and online focus group discussions (online FGDs). The study population consisted of young adults aged 20 to 35 years, who work in the Jakarta area, and live in the Jakarta Metropolitan Area, which comprises Jakarta, Bogor, Depok, Tangerang, and Bekasi. Young adults were recruited via snowball convenience sampling using messaging applications and social media networks. Data were collected from July to August 2020.

The sample size calculation for the quantitative component was made based on quarantine compliance during the 2009 influenza outbreak in Australia among those who took time off work [8]. Assuming an incomplete response rate of 10% to the online survey, the final sample size was calculated as 333.

## Ethics

This study was approved by the Research Ethics Committee, Graduate School of Medicine, the University of Tokyo (serial number: 2020087NI) and the Research and Community Engagement Ethical Committee, Faculty of Public Health, Universitas Indonesia (serial number: Ket-383/UN2.F10.D11/PPM.00.02/2020). Both ethics committees approved the digital consent as it was explained in data collection section during ethic interviews. Participation in this study was voluntary and the confidentiality was strictly maintained. Prior to answering the questions, participants were asked to sign an electronic informed consent. In addition, the participants in the qualitative interviews also provided another electronic informed consent as well as oral consent at the beginning of the interviews.

## Quantitative method

The primary outcome of this study was physical distancing compliance. Compliance was defined as people's adherence to each physical distancing measure based on Indonesia COVID-19 guidelines [20]. The level of physical distancing was evaluated as a total score from nine physical distancing measures. These nine measures were 1) maintaining a 1-meter distance, 2) avoiding handshakes, 3) avoiding hugs, 4) avoiding public transportation, 5) working/studying from home, 6) avoiding gatherings and crowds, 7) postponing meetings, 8) avoiding visiting elderly people, and 9) praying at home. For each measure, "never" was scored as 1, "sometimes" was scored as 2, and "always" was scored as 3. The predictor variables were COVID-19-related variables such as COVID-19 testing, knowledge of COVID-19, etc.; and religious and tradition-related activities such as breaking Ramadan fast together, joining *mudik* tradition, etc. *Mudik* is a traditional homecoming exodus in Indonesia that happens during Eid, the biggest Muslim holiday celebration. The question related to COVID-19 variables were adapted from the WHO European region COVID-19 survey tool and previous COVID-19 survey in the United States and the United Kingdom [21, 22]. These variables were constructed based on composite questions and scored based on its answer. For example, for the knowledge of COVID-19 variables, each question under this variable was scored "1 for true answer" and "0 for false answer". Covariate variables were sociodemographic characteristics. Details about the COVID-19-related variables and religious and tradition-related activities are shown in S1 Table.

The questionnaire was translated into Indonesian, and reviewed by three authors. The face validity and content validity were reviewed by public health experts from the Faculty of Public Health, Universitas Indonesia. Using the Google Form platform, an online self-administered questionnaire was uploaded on this study's website (http://survey-covid19.herokuapp.com/). This questionnaire was also pre-tested with 30 young adults, and after completing the pre-test, online discussions were conducted to ensure that they were able to understand the questionnaire. The questionnaire consisted of 42 questions. The reliability of the instrument in the current study was acceptable, with a Cronbach's alpha of 0.66.

Hierarchical linear regression analyses were performed to identify facilitators and barriers to physical distancing compliance. In step 1, Model 1 was adjusted for sociodemographic characteristics such as age, place of working/studying, place of living, sex, education level, occupation, resuming work at office/study at school, living situation, and monthly income/allowance.

In step 2, Model 2 was adjusted for both sociodemographic and COVID-19-related variables such as COVID-19 testing, knowledge of COVID-19, knowledge of COVID-19 prevention, perceived severity, perceived susceptibility, reasons for compliance, feasibility for compliance, duration of restriction compliance, understanding physical distancing guideline, perceived benefit, receiving workplace/school support, and receiving government support. In step 3, Model 3 was additionally adjusted for religious and tradition-related variables such as special prayer together outside during pandemic, breaking Ramadan fast together outside during pandemic, religious celebration during pandemic, and *mudik* tradition during pandemic. The variance inflation factor (VIF) values were also obtained to determine the presence of multicollinearity. The VIF values for all variables were less than 5.8 [23]. All statistical significance was set at $p < 0.05$. All statistical analyses were performed using Stata Release 13 (StataCorp. 2013. College Station, TX: Stata Press). The Strengthening the Reporting of Observational Studies in Epidemiology (STROBE) for cross-sectional assessment is reported in S1 Checklist.

## Qualitative method

At the end of the quantitative questionnaire, the young adults were asked to provide their contacts if they were interested in participating in an online FGD and photovoice. A maximum variation sampling was employed to ensure the representativeness and diversity of participants to obtain more comprehensive answers [24].

For online FGDs and photovoice, a topic guide was developed based on the quantitative questionnaire. The photovoice ethics guideline was adapted from a study conducted by Shumba et al and translated into Indonesian [25]. The content of the topic guide includes: 1) how long the participants complied with physical distancing for, 2) what kind of physical distancing measures and religious tradition activities were difficult or easy to comply during the pandemic and the reason behind it, and 3) how their photos supported their explanation of their experience on physical distancing compliance. This topic guide was also pre-tested with three and two young adults for online FGD and photovoice, respectively. Initial meetings with each young adult were conducted by phone or video call prior to the online FGDs and photovoice to explain the study's purpose and establish rapport. They also were asked to take one to three photographs that described their facilitators or barriers to physical distancing compliance. Separate online group discussions for the online FGDs and photovoice were conducted using Zoom or Google Meet. The young adults were asked to turn off their cameras during the discussion to protect their privacy. Photographs taken by the young adults were used in the online discussion for photovoice. Each group discussion lasted for approximately one hour. Data saturation is reached when no new information was obtained from additional FGDs. This was confirmed by referring to the fieldnotes and the themes in the codebook. Each participant received an incentive of a 100,000 Indonesian rupiah gift card (equivalent to US$7.00) after finishing the discussion.

All the recordings were transcribed verbatim, and the initial transcripts were coded in Indonesian. A thematic analysis was employed following the steps recommended by Nowel et al. (2017) to ensure the trustworthiness of the analysis and results [26]. The themes were generated deductively and inductively and discussed in the peer debriefing. The Consolidated Criteria for Reporting Qualitative Research (COREQ) checklist is reported in S2 Checklist.

## Integration

The results of the quantitative and qualitative analyses were integrated in joint displays. Joint displays provide a visual means for mixed methods results in the generation of new inferences

[27]. The joint displays in this study showed the quantitative result, online discussion quotes with a photograph and caption.

## Results

### Quantitative findings

The final sample for this study was 330 participants. Table 1 summarizes the sociodemographic characteristics and physical distancing scores of the young adults. Three hundred and thirty young adults participated in this study, of whom 209 (63.3%) were women; the mean age was 27.2 years (SD 3.3). Regarding their place of residence, 176 (53.5%) lived in the Jakarta area. The majority were currently working or studying in Jakarta Selatan (43.0%) and Jakarta Pusat (33.9%). Regarding their working/studying status as of July 2020, 216 (65.4%) resumed work/study at office/school and 23 (7.0%) never worked or studied from home, while 91 (27.6%) remained working/studying from home. Descriptive statistics for the other variables are shown in S1 Table.

The mean compliance score was 23.2 out of 27.0 points. As shown in Fig 1, compared with other physical distancing measures, the young adults had lower compliance for maintaining a 1-meter distance, working/studying from home, avoiding gatherings and crowds, and postponing meetings.

Table 2 shows factors associated with the physical distancing compliance score. After adjusting for sociodemographic characteristics in model 1, working in the health sector and working or studying from home were significantly associated with a higher physical distancing compliance score while never working or studying from home was associated with a lower physical distancing score. In model 2, however, after adjusting for sociodemographic and COVID-19-related variables, working in the health sector and never working/studying from home were not statistically associated with physical distancing compliance.

In model 3, after adjusting for religious and tradition-related variables, in addition to the variables in models 1 and 2, working or studying from home ($\beta = 0.14$, $p < 0.05$) was positively associated with higher physical distancing compliance scores compared to working in an office or studying at a school. Moreover, the duration of restriction compliance, such as one week to one month ($\beta = 0.24$, $p < 0.01$) and more than one month ($\beta = 0.35$, $p < 0.001$), were positively associated with higher physical distancing compliance scores compared to less than one week compliance and never complied with COVID-19 restriction. Likewise, feasibility to comply ($\beta = 0.24$, $p < 0.001$) and having a higher score on understanding the COVID-19 guidelines ($\beta = 0.21$, $p < 0.001$) were positively associated with higher physical distancing compliance scores. Participating in religious celebrations with others during the pandemic was negatively associated with higher physical distancing compliance scores ($\beta = -0.15$, $p < 0.05$) compared to celebrating only with family at home. Compared to not joining *mudik* this year, having hometown in the Jakarta Metropolitan Area was negatively associated with higher physical distancing compliance scores ($\beta = -0.12$, $p < 0.05$).

### Qualitative findings

Forty-seven young adults—30 women and 17 men—participated in the qualitative study. Among them, 18 young adults participated in five online FGDs. Data saturation was achieved within five groups. For the photovoice component, 29 young adults submitted 66 photographs. Most of them lived in the Jakarta area (n = 20, 42.5%), worked in Jakarta Selatan (n = 23, 48.9%), and had returned to work at an office and study at a school after the PSBB transition on June 5, 2020 (n = 30, 63.8%) (Table 3). Several major themes on the facilitators of and barriers to physical distancing compliance were obtained.

**Table 1. Descriptive statistics of sociodemographic characteristics and physical distancing compliance scores of 330 young adults.**

| | n | % |
|---|---|---|
| **Physical distancing score, mean [SD]** | 23.2 [2.6] | |
| **Age, mean [SD]** | 27.2 [3.3] | |
| **Sex** | | |
| Men | 121 | 36.7 |
| Women | 209 | 63.3 |
| **Place of living** | | |
| Jakarta | | |
| Jakarta Barat | 21 | 6.4 |
| Jakarta Pusat | 31 | 9.4 |
| Jakarta Selatan | 54 | 16.4 |
| Jakarta Timur | 50 | 15.2 |
| Jakarta Utara | 20 | 6.1 |
| Bogor | | |
| Kabupaten Bogor | 14 | 4.2 |
| Kota Bogor | 13 | 4.0 |
| Depok | | |
| Kota Depok | 52 | 15.8 |
| Tangerang | | |
| Kota Tangerang | 16 | 4.9 |
| Kota Tangerang Selatan | 20 | 6.0 |
| Bekasi | | |
| Kabupaten Bekasi | 8 | 2.4 |
| Kota Bekasi | 31 | 9.4 |
| **Place of working/studying** | | |
| Jakarta Pusat | 112 | 33.9 |
| Jakarta Utara | 15 | 4.5 |
| Jakarta Selatan | 142 | 43.0 |
| Jakarta Barat | 24 | 7.3 |
| Jakarta Timur | 37 | 11.2 |
| **Occupation** | | |
| Public sector | 58 | 17.6 |
| Private sector | 200 | 60.6 |
| Health sector | 18 | 5.4 |
| College student | 23 | 7.0 |
| Others | 31 | 9.4 |
| **Education level** | | |
| High school | 23 | 7.0 |
| Diploma | 35 | 10.6 |
| University | 236 | 71.5 |
| Graduate and Postgraduate | 36 | 10.9 |
| **Monthly income/allowance before pandemic in Indonesia Rupiah (US dollar)** | | |
| Less than 3 million (US$212) | 31 | 9.4 |
| 3 to 6 million (US$212 to US$424) | 112 | 33.9 |
| 6 to 9 million (US$424 to US$637) | 90 | 27.3 |
| More than 9 million (US$637) | 97 | 29.4 |

(*Continued*)

**Table 1.** (Continued)

| | | |
|---|---|---|
| **Physical distancing score, mean [SD]** | **23.2 [2.6]** | |
| **Age, mean [SD]** | **27.2 [3.3]** | |
| | **n** | **%** |
| **Living situation** | | |
| Alone | 30 | 9.1 |
| With one person or more | 300 | 90.9 |
| **Resuming work at office/study at school (as July 21, 2020)** | | |
| Yes | 216 | 65.4 |
| Working/studying from home | 91 | 27.6 |
| Never worked/studied from home | 23 | 7.0 |

**Implementation of the government policy.**   The first theme described a barrier to comply with physical distancing measures. Young adults expressed their views on the implementation of the government policy. The implementation hindered their compliance with physical distancing measures.

"…*according to company regulation where I work and regulation from uh.. the government also allows (the company) to work from the office..*"

–Man, 28 years old, State-owned Enterprises employee, Kota Depok, Photovoice.

**Having social pressure.**   The second theme also described a barrier. Young adults also expressed their uneasiness about complying with physical distancing because it could go against social norms.

"…*in my big family, I heard a bad talk. My family's intention was not to come to (the religious celebration) or to come but after a few days to avoid a mass gathering. However, there was a bad talk in a big family. So, we kept coming…*"

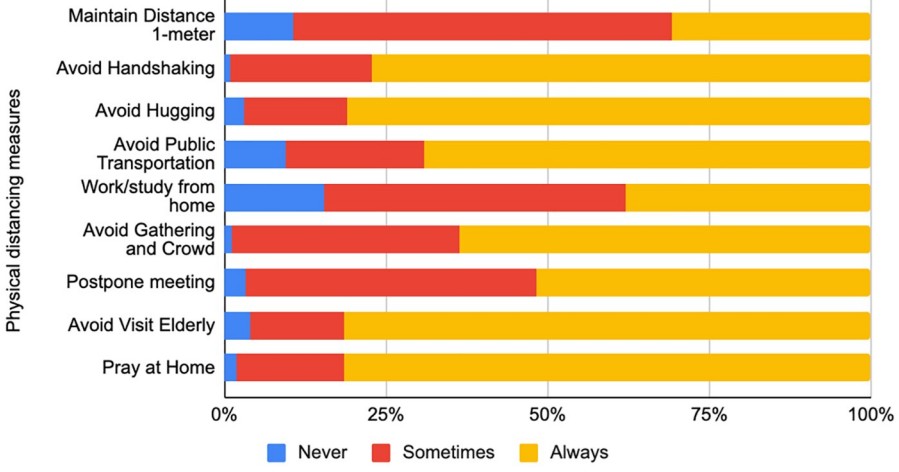

**Fig 1. A graph of compliance among young adults for each physical distancing measure.**

**Table 2. Hierarchical linear regression analysis with physical distancing compliance scores.**

| Covariate variables | Model 1 $\beta$ | Model 2 $\beta$ | Model 3 $\beta$ |
|---|---|---|---|
| **Occupation** (vs Public Sector) | | | |
| Private sector | -0.13 | -0.11 | -0.10 |
| Health sector | **0.11**\* | 0.05 | 0.05 |
| College student | -0.14 | -0.07 | -0.06 |
| Others | -0.04 | -0.02 | 0.02 |
| **Resuming work at office/study at school (as of July 21, 2020)** (vs. Yes) | | | |
| Working/studying from home | **0.20**\*\* | **0.14**\* | **0.14**\* |
| Never worked/studied from home | **−0.12**\* | −0.06 | −0.08 |
| **Feasibility to comply with physical distancing score** | | **0.23**\*\*\* | **0.24**\*\*\* |
| **Duration of restriction compliance** (vs. Never and less than 1 week) | | | |
| 1 week to 1 month | | **0.26**\*\* | **0.24**\*\* |
| More than 1 month | | **0.39**\*\*\* | **0.35**\*\*\* |
| **Understanding physical distancing guideline score** | | **0.27**\*\*\* | **0.21**\*\*\* |
| **Religious celebration during pandemic** (vs. No, at home with family only) | | | |
| Yes | | | **−0.15**\* |
| Never celebrate and did not join celebration with others | | | 0.13 |
| ***Mudik* tradition during pandemic** (vs. Did not join *mudik* this year) | | | |
| Yes | | | 0.04 |
| Living in the Jakarta Metropolitan Area | | | **−0.12**\* |
| $R^2$ | 0.15 | 0.40 | 0.45 |
| $\Delta R^2$ | 0.15 | **0.25**\*\*\* | **0.05**\*\* |

$R^2$ = variance; $\Delta R^2$ = change in variance. Statistical significance indicated by $p < 0.05$\*; $p < 0.01$\*\*; $p < 0.001$\*\*\*.

Model 1 was adjusted for sociodemographic characteristics.

Model 2 was adjusted for sociodemographic characteristics and COVID-19-related variables.

Model 3 was adjusted for sociodemographic characteristics, COVID-19-related variables, and religious and tradition-related variables. Table 2 only shows significant variables; the full version of this table is available in S2 Table.

–Woman, 29 years old, project management officer, Kota Tangerang, Photovoice.

"...honestly, even though I refused (to shake hands), but sometimes our Eastern culture (makes me) feel guilty if someone comes to us and......demands to shake hands. So, in the end, I did handshake, whereas we (I) were reluctant..."

–Woman, 25 years old, civil servant, Kota Tangerang, Online FGD.

**Having clear and easy-to-understand indicators of physical distancing at public facilities.** The third theme described a facilitator to compliance. One of the young adults mentioned the ease of maintaining distance during Friday prayer in a mosque. Another young adult also shared her experience at a coffee shop that had clear indicators to help her to comply with physical distancing measures.

"On Friday, I saw...the mosque behind my office, performing Friday prayer, and I looked closely. They maintain distance..."

–Woman, 28 years old, private employee, Kota Tangerang Selatan, Photovoice.

**Table 3. Sociodemographic characteristics of young adults in the qualitative phase (*n* = 47).**

| | FGD (*n* = 18) | Photovoice (*n* = 29) | Total (*n* = 47) |
|---|---|---|---|
| **Age, mean [SD]** | 26.8 [3.6] | 26.8 [2.8] | |
| | n | n | n (%) |
| **Sex** | | | |
| Men | 9 | 8 | 17 (36.2) |
| Women | 9 | 21 | 30 (63.8) |
| **Place of living** | | | |
| Jakarta | 7 | 13 | 20 (42.5) |
| Bogor | 2 | 1 | 3 (6.4) |
| Depok | 3 | 3 | 6 (12.8) |
| Tangerang | 5 | 6 | 11 (23.4) |
| Bekasi | 1 | 6 | 7 (14.9) |
| **Place of working/studying** | | | |
| Jakarta Pusat | 7 | 10 | 17 (36.2) |
| Jakarta Utara | 1 | 0 | 1 (2.1) |
| Jakarta Selatan | 8 | 15 | 23 (48.9) |
| Jakarta Barat | 1 | 2 | 3 (6.4) |
| Jakarta Timur | 1 | 2 | 3 (6.4) |
| **Occupation** | | | |
| Public sector | 2 | 5 | 7 (14.9) |
| Private sector | 9 | 16 | 25 (53.2) |
| Health sector | 2 | 3 | 5 (10.6) |
| College student | 2 | 2 | 4 (8.5) |
| Others | 3 | 3 | 6 (12.8) |
| **Education level** | | | |
| High school | 0 | 2 | 2 (4.3) |
| Diploma | 2 | 3 | 5 (10.6) |
| University | 13 | 22 | 35 (74.5) |
| Graduate and Postgraduate | 3 | 2 | 5 (10.6) |
| **Monthly income/allowance before pandemic in Indonesian rupiah** | | | |
| Less than 3 million (US$212) | 3 | 1 | 4 (8.5) |
| 3 to 6 million (US$212 to US$424) | 4 | 9 | 13 (27.7) |
| 6 to 9 million (US$424 to US$637) | 4 | 11 | 15 (31.9) |
| More than 9 million (US$637) | 7 | 8 | 15 (31.9) |
| **Living situation** | | | |
| Alone | 1 | 2 | 3 (6.4) |
| With one person or more | 17 | 27 | 44 (93.6) |
| **Resuming work at office/study at school (as July 21, 2020)** | | | |
| Yes | 14 | 16 | 30 (63.8) |
| Working/studying from home | 3 | 8 | 11 (23.4) |
| Never worked/studied from home | 1 | 5 | 6 (12.8) |

"*We can still buy a (cup of) coffee in a coffee shop and still feel safe because the cashier wears a mask, for the customer, there was. . . a transparent curtain; then for payment, we were suggested to use online payment such as Gopay (e-money) or debit card to avoid physical contact.*"

–Woman, 29 years old, private employee, Kota Tangerang, Photovoice.

**Workplace policies and supports.**   Young adults also explained that having support from their workplace encouraged their compliance. However, a young adult who work in informal sector had difficulties to comply with physical distancing measures. Workplace policies and supports can be both facilitator and barrier.

"...*So, one week work from home, the week after at office. (Only) 50% of the staff come to the office. That is the policy of my workplace to promote physical distancing.*"

–Woman, 29 years old, private employee, Kota Tangerang, Photovoice.

"...*Even before the pandemic, my office already had implemented the Work from Home (WFH) policy once a week, so during pandemic, it was just like extending the duration of WFH (from once a week to more often) and it did not affect to my work.*"

–Woman, 29 years old, project management officer, Kota Tangerang, Photovoice.

"*I just quit the company in February 2020, so for the workplace policy I didn't know very well, to be honest I just started a business.. ... .because I just started my business so I need to meet people, so I had difficulty to maintain distance, uh.. to avoid to meet with others...*"

–Man, 23 years old, entrepreneur, Kota Tangerang Selatan, Online FGD.

**Having self-awareness and awareness inspired by the actions of others.**   Young adults discussed how awareness, or the lack thereof, could be a facilitator of or a barrier to compliance with physical distancing. Some also discussed that their awareness was inspired by other people's actions.

"...*Because of staff awareness and maybe also their fear, if the elevator is already full of only four people, we let them go first.*"

–Man, 28 years old, State-Owned Enterprises employee, Jakarta Timur, Photovoice.

"...*to keep uh.. avoiding crowds that I think is not essential or not important, such as coming to gathering to hangout or arisan (regular social gathering)...*"

–Man, 31 years old, dentist, Jakarta Selatan, Online FGD.

"*Some people were concerned about health protocol but the rest of them seemed ignorant.. .. It can be said that if there was a staff member who encourages people to follow the rules, they would do . . . but if not, they would be ignorant and do not care.*"

–Woman, 25 years old, private employee, Jakarta Barat, Photovoice.

## Integration

Three major findings were obtained by integrating the quantitative and qualitative results. In the first joint display, the photo shows young adults using a video communication software (Zoom) to work from home (Fig 2). Quantitative result then showed that working or studying from home was positively associated with higher compliance scores. In the qualitative part, a young adult explained how her workplace policy supported physical distancing measures by

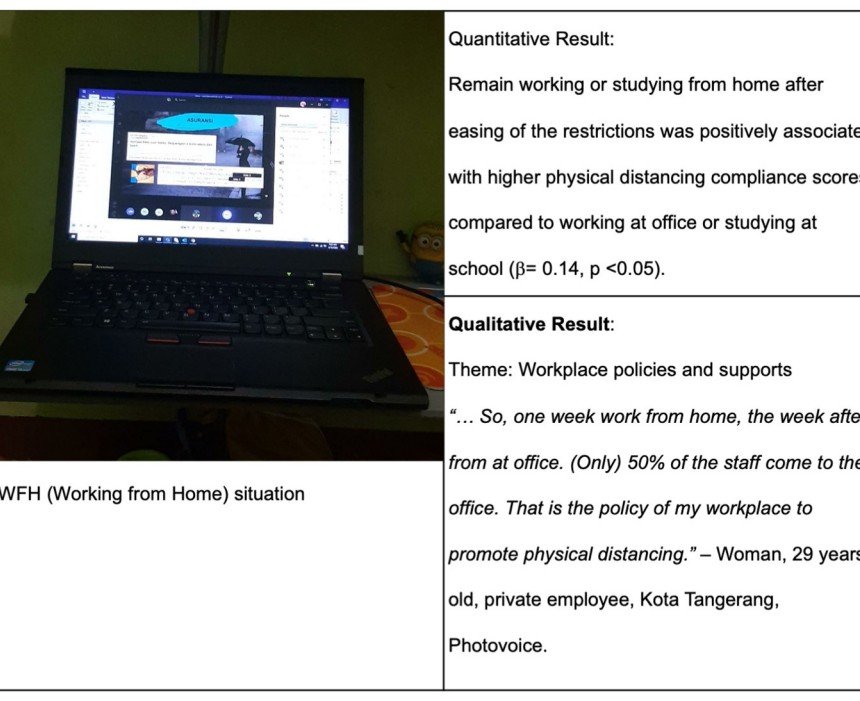

**Fig 2. Joint display of workplace and government policies, awareness, and physical distancing compliance.**

allowing staff to work from home. Another young adult mentioned that after the government's easing of restrictions, his workplace required the staff to resume working from the office.

The next joint display (Fig 3) shows a photo of people following the guideline to maintain distance in the elevator. In the quantitative result, having good understanding of the guidelines was positively associated with higher compliance. From the qualitative result, young adults mentioned that some actions that led to non-compliance, such as using crowded elevator and joining social gathering, should be avoided, as written in the guidelines.

A photo in the final integrated finding (Fig 4) shows people maintaining distance during Friday prayer at a mosque. However, quantitative result showed that religious celebrations were negatively associated with higher compliance. Qualitative result then explained its reasons. A young adult, for example, explained that she felt guilty to refuse to handshake (or "Salam") because it is part of a religious custom even though she knows handshaking during pandemic should be avoided.

## Discussion

This study has three major findings. First, identified facilitators for better compliance were working or studying from home and better awareness and flexible workplace policies. Second, its barriers were celebrating the religious day, having hometown in the Jakarta Metropolitan Area, resuming work at the office, and social pressure to celebrate religious days with family. Third, compliance with overall physical distancing measures was high among young adults.

Young adults took appropriate actions to protect themselves from COVID-19 infection and this was reflected by the high compliance with physical distancing measures in this study. The

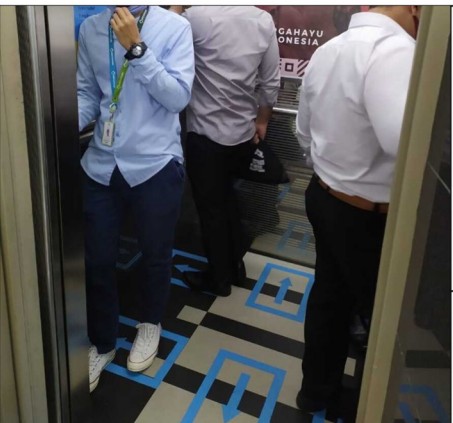

**Quantitative Result:**

Having a higher score on understanding the guidelines was positively associated with higher physical distancing compliance ($\beta$= 0.21, p <0.001).

Physical distancing at an office elevator.

**Qualitative Result:**

Theme: Having self-awareness and awareness inspired by the actions of others

*"...Because of staff awareness and maybe also their fear, if the elevator is already full of only four people, we let them go first."* – Man, 28 years old, State-Owned Enterprises employee, Jakarta Timur, Photovoice.

Meta-Inference: Understanding the guideline was an essential driver in increasing the awareness to comply with physical distancing.

**Fig 3. Joint display of understanding guideline, awareness, and physical distancing compliance.**

actions were facilitated by flexible workplace policies. Employee protection policies encourage physical distancing compliance among young adults. Worker financial protection policies, such as guaranteeing job security and income replacement, successfully promoted voluntary quarantine compliance in Canada during the SARS outbreak [28]. Worker health protection policies, such as allowing workers to take time off work and to remain working from home, were also associated with compliance with recommendation measures [8, 29]. Remote working, however, was only applicable to those who work in the formal sector. People who work in informal sectors with unstable income might not comply with physical distancing measures. Moreover, this study found that becoming an entrepreneur was an alternative after losing a job due to the pandemic. Consequently, hindering physical distancing compliance due to the need to start a business.

Having awareness fostered physical distancing compliance among young adults—being aware of what is going on around them increased their physical distancing compliance [30]. In this study, young adults were mindful of keeping themselves away from crowds, including avoiding unimportant gatherings. Awareness might be increased if physical distancing indicators at public facilities were clear and easy to understand. COVID-19 clusters in public facilities showed that activities held in public facilities are not completely safe [31, 32]. Poor public awareness was one of the barriers to physical distancing, especially when using mass transportation. Even though young adults generally adhere to the physical distancing measures, however they cannot influence other people's behavior. The level of public awareness influences physical distancing compliance [33]; for example, people using mass transportation might not be able to adhere to the recommended safeguard measures [34]. Having a better

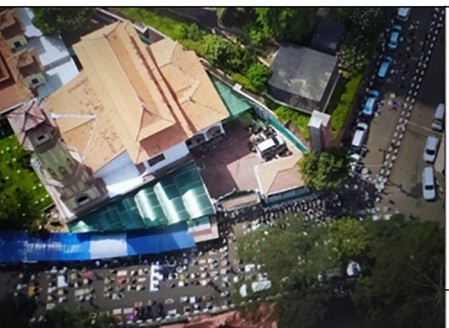

**Quantitative Result:**

Celebrating the religious day with others during the pandemic was negatively associated with higher physical distancing compliance scores (β= −0.14, p <0.05) compared to celebrating only with family at home.

New Normal: Friday prayer (maintain) distance

**Qualitative Result:**

Theme: Having social pressure

"...honestly even though I refused (to shake hands), but sometimes our eastern culture (makes me) feel guilty if someone comes to us and ... demands to shake hands. So in the end, I did handshake, whereas we (I) were reluctant ..."– Woman, 25 years old, civil servant, Kota Tangerang, Online FGD.

Meta-Inference: Social pressures, including norms, played a profound role as a hindrance to complying with physical distancing.

**Fig 4. Joint display of clear indicators, social pressure, and physical distancing compliance.**

understanding of the COVID-19 guidelines was associated with higher physical distancing compliance. This finding also aligns with Webster et al. (2020) who reported that greater knowledge of quarantine protocols was associated with higher adherence to quarantine or isolation in previous studies [10]. In a study in Sierra Leone during the Ebola outbreak, lack of understanding of the term 'isolation' led to non-compliance with quarantine measures [35]. The feasibility of adhering to physical distancing measures was also revealed as one of the factors associated with compliance. The ease of performing each physical distancing measure encouraged compliance. When people had more pressing concerns, such as working or seeking medical treatment, they tended to ignore the rules [36–38].

Even though preventive actions were performed by young adults, however, they still found some physical distancing measures difficult to comply, such as maintaining a 1-meter distance, working/studying from home, avoiding gatherings, and postponing meetings with others. This finding was similar to a national survey on community behavior in Indonesia during the COVID-19 pandemic [39]. They tended to be driven by several reasons such as cultural and religious events and ad-hoc policies by the workplace and government. Social pressure was found to be a major barrier to physical distancing compliance. It evoked a guilty feeling to comply with physical distancing measures. Compliance with these measures was in contrast with societal values, especially related to religion and tradition. As a result, social conformity led them to non-compliance [40]. This study found that religious celebration and having hometown in the Jakarta Metropolitan Area were associated with lower compliance scores. At least once a year, Indonesians will do *mudik* to gather with family and celebrate traditional religious activities, especially during Ramadan and Eid. Many Indonesians feel obliged to

celebrate religious days with family and relatives. Consequently, the religious celebration provided the possibility of breaking physical distancing measures. Sociocultural factors, including social norms, play a vital part in increasing and reducing quarantine adherence [10]. Moreover, social norms were found to be one of the determinants of an individuals' intention to comply with physical distancing [41].

Young adults tended to be steered by ad-hoc policies by the workplace and the government. The Work-at-Office policy introduced after the government lifted the restriction, acted as a barrier to physical distancing compliance. Young adults found it difficult to comply with physical distancing when they had to return to work at the office. This finding was consistent with the previous studies in North America and Europe [29, 42]. People who resumed their work at the office/study at school during the transition restriction did not intend to comply with physical distancing measures. Among Chinese factory workers, their compliance was relatively low [43]. They were unable to avoid social and meal gathering, and crowded places at the beginning of work resumption [43]. Testing and contact tracing were important actions to prevent the next epidemic wave before resuming usual activities [44]. The number of daily COVID-19 tests in Indonesia, however, is low compared to neighboring countries [45]. After easing the PSBB on June 5, 2020, a model from Ariawan et al. (2020) showed a reduction in the number of people staying home and an increase in estimated cases in Jakarta [46]. A longer duration of restriction compliance was positively associated with higher physical distancing scores in this study. However, remaining at home for an extended period affected psychological feelings such as boredom. Consequently, even though people remained working from home after the restriction was lifted, they might be less willing to comply if the duration is prolonged.

To the best of our knowledge, this study is the first to focus on physical distancing compliance and its facilitators and barriers using a combination of an online questionnaire, online FGDs, and photovoice. Combining the three methods enabled the researchers to gain multiple insights into the situation. Photovoice empowered the young adults to be more aware of their surroundings and to identify and represent the issues in their neighborhoods. Photovoice also provides participants' perspective for stakeholders and policy makers to create guidelines to end the current pandemic. The analysis and findings were enriched by the current mixed methods design, in which quantitative and qualitative findings were integrated.

This study has several limitations. First, online recruitment through social media could only target active users which might have selection bias. Second, nonverbal communications could not be incorporated because participants were asked to turn off their cameras during the online group discussions to protect their privacy, especially of COVID-19 patients. The online recruitment method and data collection, however, were the safest methods during the ongoing COVID-19 pandemic in Indonesia and elsewhere.

## Conclusions

Young adults took appropriate physical distance to protect themselves from COVID-19 infection. Their better physical distancing compliance was facilitated by flexible workplace policies and better awareness. However, it was only applicable to young adults who work in the formal sector. Even though the physical distancing compliance score was high, they still tended to be driven by cultural and religious events and ad-hoc policies by the workplace.

All stakeholders should continue to collaborate to take effective actions to improve physical distancing. The national government should continue to involve the Indonesian Ulema Council (Majelis Ulama Indonesia) and other religious councils to encourage religious leaders to spread awareness in the community through the fatwas (legal opinion or ruling issued by an

Islamic scholar) and religious teaching. Moreover, companies and workplaces should protect staff safety by creating and implementing health protocols at work strictly.

## Supporting information

**S1 Checklist. The Strengthening the Reporting of Observational Studies in Epidemiology (STROBE) for cross-sectional assessment.**
(DOC)

**S2 Checklist. The Consolidated Criteria for Reporting Qualitative Research (COREQ) checklist.**
(DOCX)

**S1 Table. Descriptive statistics of COVID-19 related variables of 330 young adults.**
(DOCX)

**S2 Table. Hierarchical linear regression analysis with physical distancing compliance score (full version).**
(DOCX)

**S1 Questionnaire. Compliance questionnaire (English).**
(DOCX)

**S2 Questionnaire. Compliance questionnaire (Indonesian).**
(DOCX)

**S1 Interview guide. Photovoice interview guide (English).**
(DOCX)

**S2 Interview guide. Photovoice interview guide (Indonesian).**
(DOCX)

**S3 Interview guide. Online FGDs interview guide (English).**
(DOCX)

**S4 Interview guide. Online FGDs interview guide (Indonesian).**
(DOCX)

## Acknowledgments

We are grateful to all the participants for their contributions to this study. In particular, we thank Futuhal Arifin Annasri for creating our study website for data collection, and the authors' social network for helping to share our study website during data collection. We would like to thank Editage (www.editage.com) for English language editing.

## Author Contributions

**Conceptualization:** Ahmad Junaedi, Ken Ing Cherng Ong.

**Data curation:** Ahmad Junaedi, Ken Ing Cherng Ong, Fauzan Rachmatullah.

**Formal analysis:** Ahmad Junaedi, Ken Ing Cherng Ong, Akira Shibanuma.

**Funding acquisition:** Masamine Jimba.

**Investigation:** Ahmad Junaedi, Ken Ing Cherng Ong, Fauzan Rachmatullah.

**Methodology:** Ahmad Junaedi, Ken Ing Cherng Ong, Akira Shibanuma, Junko Kiriya, Masamine Jimba.

**Project administration:** Ahmad Junaedi, Fauzan Rachmatullah.

**Resources:** Ahmad Junaedi.

**Software:** Ahmad Junaedi, Akira Shibanuma.

**Supervision:** Ken Ing Cherng Ong, Masamine Jimba.

**Validation:** Ahmad Junaedi, Ken Ing Cherng Ong.

**Visualization:** Ahmad Junaedi, Ken Ing Cherng Ong, Akira Shibanuma, Junko Kiriya, Masamine Jimba.

**Writing – original draft:** Ahmad Junaedi, Ken Ing Cherng Ong.

**Writing – review & editing:** Ahmad Junaedi, Ken Ing Cherng Ong, Fauzan Rachmatullah, Akira Shibanuma, Junko Kiriya, Masamine Jimba.

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
