## [Decision Letter · Decision Letter 0]

9 Aug 2021

 PGPH-D-21-00070 Factors influencing physical distancing compliance among young adults during COVID-19 pandemic in Indonesia: A photovoice mixed methods study PLOS Global Public Health

Dear Dr. Ong,

Thank you for submitting your manuscript to PLOS Global Public Health. After careful consideration, we feel that it has merit but does not fully meet PLOS Global Public Health’s publication criteria as it currently stands. Therefore, we invite you to submit a revised version of the manuscript that addresses the points raised during the review process.

We look forward to receiving your revised manuscript.

Kind regards,

Daniel Kim, M.D., Dr.P.H.

Academic Editor

Journal Requirements:

Additional Editor Comments (if provided):

Reviewers' comments:

Reviewer's Responses to Questions

**Comments to the Author**

1. Does this manuscript meet PLOS Global Public Health’s publication criteria? Is the manuscript technically sound, and do the data support the conclusions? The manuscript must describe methodologically and ethically rigorous research with conclusions that are appropriately drawn based on the data presented.

Reviewer #1: Yes

Reviewer #2: Partly

2. Has the statistical analysis been performed appropriately and rigorously?

Reviewer #1: I don't know

Reviewer #2: I don't know

3. Have the authors made all data underlying the findings in their manuscript fully available (please refer to the Data Availability Statement at the start of the manuscript PDF file)?

Reviewer #1: Yes

Reviewer #2: Yes

4. Is the manuscript presented in an intelligible fashion and written in standard English?

Reviewer #1: Yes

Reviewer #2: Yes

5. Review Comments to the Author

Reviewer #1: The manuscript explores physical distancing compliance and its facilitators and barriers among Indonesian young adults aged 20–35 years, using a combination of an online questionnaire, online focus group discussions, and photovoice. I find the paper timely, and it has the potential to make a valuable contribution to the existing literature on the topic. However, I have some comments I would like the authors to consider.

Abstract:

1.) Please consider clarifying the first sentence of the abstract, similarly to the beginning of the Introduction section. When was the spreading of the coronavirus disease out of control?

Introduction:

2.) “These studies showed that several factors were associated with quarantine adherence such as social and psychological factors, knowledge of the disease, value…”

What kind of value? Please clarify the sentence.

Methods:

3.) Why did the authors decide to focus on the population aged 20-35 years? Other age groups were also heavily affected by the pandemic, and the workforce includes other age groups as well. Please consider explaining this choice in the text. Also, this could lead to issues with representativeness since the driving force behind the behavior – in this case, physical distancing compliance – of young adults might differ from that of, e.g., older adults. Please consider discussing this among the other limitations.

4.) “The level of physical distancing was evaluated as a total score from nine physical distancing measures where a higher score indicates higher compliance.”

What were the potential answers to the physical distancing measure questions? How were the answers scored? What is the total score? The total score is first mentioned in the results section, while Figure 1 shows the possible answers to the questions. However, this should be explained as part of the methodology.

5.) “The exposure variables in this study were sociodemographic characteristics; COVID-19-related variables such as COVID-19 testing, knowledge of COVID-19, etc.; and religious and tradition-related activities such as breaking fast during Ramadan, joining Mudik tradition, etc.”

“Model 1 was adjusted for sociodemographic characteristics. Model 2 was adjusted for both sociodemographic and COVID-19-related variables. Model 3 was additionally adjusted for religious and tradition-related variables.”

Please make a detailed list of the main exposure variables and covariates in the text. Consider creating a separate sub-section (or sub-sections) to provide this information. It should be clear which variables were included in each model, so categorizing the factors (e.g., sociodemographic characteristics, COVID-19-related variables, and religious and tradition-related variables) when listed might be reasonable as well.

6.) Please explain how the hierarchical models were defined. E.g., is this a two-level model where individuals are nested within living areas?

7.) Please consider including a short explanation of the content of the topic guide for online FGDs, maybe as supplementary material.

Results:

8.) “A graph of physical distancing compliance among young adults for each measure”

What type of measures are shown on the graph? Please clarify the Figure title.

9.) “As shown in Fig 1, the young adults had lower compliance for maintaining a 1-meter distance, working/studying from home, avoiding gatherings and crowds, and postponing meetings.”

Lower compliance than what? Also, what does it mean that it was lower? Please consider quantifying the presented results (e.g., x% of the participants never maintained a distance of 1-meter from others).

10.) Please consider describing the Table 2 Model 1 results using actual effect estimates in the text, similarly to the description of Model 3.

11.) Please consider including a short explanation or footnote on the Mudik tradition for the sake of international readers, especially since it is used as a separate risk factor from “religious celebration during pandemic”. It is first mentioned in the Discussion section in more detail, but an earlier explanation – maybe in the Methods section – might help readers understand the main results.

12.) In Table 3, please consider reporting percentages as well – consistent with Table 1.

13.) Please consider using actual numbers and percentages to describe the results presented in Table 3 – similarly to Table 1 results.

14.) Grammatical errors in the quoted texts of the qualitative findings are intended to reflect the style of the original Indonesian voice recording (e.g., “did not affect to my work”)? If not, please carefully correct the translated text.

15.) Figure 3: “Understanding the content of the physical distancing guideline was an essential driver in increasing the awareness to comply with physical distancing.”

In the results section, the following statement could be found: “From the qualitative result, young adults mentioned that some actions that led to non-compliance, such as using crowded elevator and joining social gathering, should be avoided, as written in the guidelines.” If the FGD participants mentioned the guidelines, a more fitting quote could be used for Figure 3 under the “qualitative results”. Otherwise, self-awareness and fear might not necessarily be related to the guidelines.

Discussion:

16.) “Second, its barriers were celebrating the religious day, having hometown in the Jakarta Metropolitan Area, resuming work…”

I believe here the authors meant “resuming work at the office”. Please clarify the sentence.

17.) “Moreover, this study found that becoming an entrepreneur was an alternative after losing a job due to the pandemic. Consequently, hindering physical distancing compliance due to the need to start a business.”

This is first mentioned in the Discussion section. If this is an important finding of the qualitative part of the study, please consider explaining it first in the Results section with the relevant quotes.

18.) Similarly, a difference in physical distancing compliance between those who work in the formal vs. the informal sector was first introduced in the Discussion section. Please consider explaining this as well in the Results section with the relevant quotes.

19.) Snowball sampling has several limitations, from sampling bias to issues with representativeness. Please consider discussing this among the limitations.

Reviewer #2: I have also included all of my comments in the attachment.

Manuscript title: Factors influencing physical distancing compliance among young adults during COVID-19 pandemic in Indonesia: A photovoice mixed methods study

This appears to be a well-designed study investigating the predictors of physical distancing among young adults in Indonesia. The study used a mixed method design and adequate sample size. The study is relevant to the management of the ongoing COVID-19 pandemic and could be a good fit for PlosONE Public Health. The overall value of the study, however, ca only be assessed after the reporting of the methods and results is done in a more comprehensive manner. There is some incomplete/selective reporting whereby only the significant predictors of several regression models were included in tables, and it is unclear exactly how many variables were included, and in which sequence. My recommendations are focused on improving the reporting of the methods, particularly the selection of predictor and outcome variables and the description of the analyses. All these revisions are required before a full review of this paper can be done.

Reviewer’s Comments

Introduction

1) Line 76: “Moreover, various cultural and religious activities take place from time to time” – authors might want to clarify/justify the relevance of these activities to the scope of the study.

Methods

2) I suggest including all “exposure variables” and stating clearly how they were scored

3) I suggest including the scoring of the main outcome measure, physical distancing

4) I think it is very important that the authors differentiate between predictor, covariate (currently called exposure variables, I believe incorrectly) and outcome measures- and at a minimum state what construct each variable is measuring and how it was scored and used in analyses.

5) For the quantitative data analysis – I recommend authors state clearly which variables were included in each model, in what order. It is unclear from the method (and also from the results) how many blocks/steps were included in each regression model.

Eg, In model one, we enter variables XX, YY, and ZZ in step 1, AA, BB, CC in step 2, and (if relevant) AA and BB in step 3.

Results

6) Paragraph above Table 2 (lines 177-183) – it appears that one of predictors included in Model 2, i.e., working in the health sector, was never mentioned in the method. Another predictor for the same model, never working/studying from home, it looks like this item is one of the 9 items/domains included in the outcome measure. Using the same items as both predictor and outcome is not an appropriate approach. Authors should clarify in the method which measures were used as predictors and which were used as outcomes (see comment 5 above).

Quantitative Results

7) In several parts of the results (quantitative), there is information pertaining to the method which should be deleted from the results and added to the method, data analysis section.

Eg., line 193: “In model 3, after adjusting for religious and tradition-related variables, in addition to the variables in models 1 and 2..”…this part should be clearly described in the method, in data analysis for Model 3, in the results, authors should simply report the results of the regression.

8) Table 2 – authors should report all variables included in the regression with the respective Beta coefficients and some other measure of significance as opposed to reporting solely the statistically significant predictors. This is a serious error in reporting and should be necessarily fixed!!!

9) Some of the measures in Table 2 were never described in the method. Authors should describe all measures/constructs used in analyses in the method section.

Qualitative Results

10) Line 209 authors mention data saturation. It would be good to include the criteria to establish data saturation in the method, qualitative data analysis section.

11) Qualitative findings – In general, I recommend including more detail and contextualization of each theme to make it more meaningful for the reader. Overall for all themes, it would be useful to phrase the name of the theme to reflect a barrier or a facilitator (or both, if relevant to their data) to physical distancing. The first theme appears to be a barrier whereas the second appears to be a facilitator.

12) While the citations are great to include to support the claims made about the topic, I recommend authors conduct a more in-depth analysis and at a minimum try to summarize what the participants are saying and then support their argument/claims with the qualitative data.

Adina Coroiu, PhD

Harvard TH Chan School of Public Health

6. PLOS authors have the option to publish the peer review history of their article (what does this mean?). If published, this will include your full peer review and any attached files.

**Do you want your identity to be public for this peer review?** For information about this choice, including consent withdrawal, please see our Privacy Policy.

Reviewer #1: No

Reviewer #2: **Yes: **Adina Coroiu, PhD

Harvard TH Chan School of Public Health

---

## [Decision Letter · Decision Letter 1]

7 Dec 2021

Factors influencing physical distancing compliance among young adults during COVID-19 pandemic in Indonesia: A photovoice mixed methods study

PGPH-D-21-00070R1

Dear Dr. Ong,

We're pleased to inform you that your manuscript has been judged scientifically suitable for publication and will be formally accepted for publication once it meets all outstanding technical requirements.

Within one week, you'll receive an e-mail detailing the required amendments. When these have been addressed, you'll receive a formal acceptance letter and your manuscript will be scheduled for publication.

An invoice for payment will follow shortly after the formal acceptance. To ensure an efficient process, please log into Editorial Manager at https://www.editorialmanager.com/pgph/ click the 'Update My Information' link at the top of the page, and double check that your user information is up-to-date. If you have any billing related questions, please contact our Author Billing department directly at authorbilling@plos.org.

Kind regards,

Daniel Kim, M.D., Dr.P.H.

Academic Editor

Reviewers' comments:

Reviewer's Responses to Questions

**Comments to the Author**

1. If the authors have adequately addressed your comments raised in a previous round of review and you feel that this manuscript is now acceptable for publication, you may indicate that here to bypass the “Comments to the Author” section, enter your conflict of interest statement in the “Confidential to Editor” section, and submit your "Accept" recommendation.

Reviewer #1: All comments have been addressed

2. Does this manuscript meet PLOS Global Public Health’s publication criteria? Is the manuscript technically sound, and do the data support the conclusions? The manuscript must describe methodologically and ethically rigorous research with conclusions that are appropriately drawn based on the data presented.

Reviewer #1: Yes

3. Has the statistical analysis been performed appropriately and rigorously?

Reviewer #1: Yes

4. Have the authors made all data underlying the findings in their manuscript fully available (please refer to the Data Availability Statement at the start of the manuscript PDF file)?

Reviewer #1: Yes

5. Is the manuscript presented in an intelligible fashion and written in standard English?

Reviewer #1: Yes

6. Review Comments to the Author

Reviewer #1: (No Response)

7. PLOS authors have the option to publish the peer review history of their article (what does this mean?). If published, this will include your full peer review and any attached files.

**Do you want your identity to be public for this peer review?** For information about this choice, including consent withdrawal, please see our Privacy Policy.

Reviewer #1: No
